# Data Measurements for Decentralized Data Markets

## Abstract

Decentralized data markets can provide more equitable forms of data acquisition for machine learning. However, to realize practical marketplaces, efficient techniques for seller selection need to be developed. We propose and benchmark federated data measurements to allow a data buyer to find sellers with relevant and diverse datasets. Diversity and relevance measures enable a buyer to make relative comparisons between sellers without requiring intermediate brokers and training task-dependent models.

## 1   Introduction

Massive training datasets have proved foundational to AI breakthroughs, from earlier deep learning breakthroughs in computer vision to large language models (LLM) [65, 35]. However, AI companies face increasing scrutiny and backlash for their data collection practices, resulting in lawsuits from data owners such as artists, software developers, and journalists [24, 61, 60]. As AI applications continue to be developed and deployed, more equitable and transparent means of data acquisition must be designed and implemented [53, 16]. Recently, data markets have been proposed to incentivize greater data sharing and access for data-restricted domains [9, 2]. As the ethical challenges and legal risks of acquiring data increase, data market platforms will be crucial to address the ethical and economic challenges in training AI models.

To facilitate practical data market platforms, we investigate the challenge of *seller selection* for a data buyer using a framework based on federated data measurements. We benchmark several proposed heuristic measures of *diversity* and *relevance*, which can be used by the buyer to compare the relative value of different sellers. The advantage of this federated data measurement framework is that it does not require direct access to the seller's data, is training-free, and is task-agnostic. These attributes are desirable for a decentralized marketplace to enable scalable seller selection for many different buyers. The three main steps of the data measurement framework are depicted in Figure 1. We evaluate several definitions of diversity and relevance on multiple computer vision datasets by benchmarkiing each data measurement for its ability to rank sellers, correlation with classification performance, and robustness to duplicate and noisy data. In summary, we show that federated data measurements allow private and lightweight seller discovery that can lower search costs for a data buyer in a decentralized data marketplace.

## 2   Decentralized Data Markets

Current data brokers are highly centralized and aggregate vast amounts of data, often without a user's knowledge, consent, or compensation [57, 13]. This massive centralization of data has led to increased

Submitted to the 38th Conference on Neural Information Processing Systems (NeurIPS 2024) Track on Datasets and Benchmarks. Do not distribute.

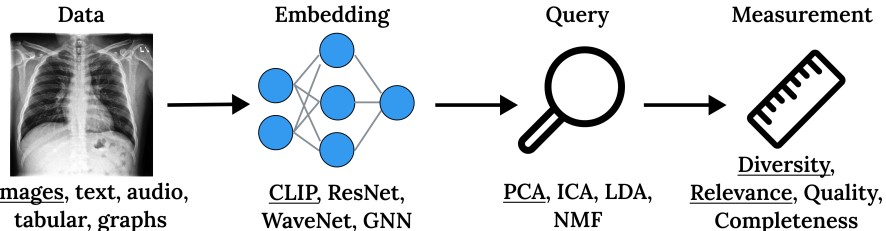

| Data | Embedding | Query | Measurement |

images, text, audio, tabular, graphs · CLIP, ResNet, WaveNet, GNN · PCA, ICA, LDA, NMF · Diversity, Relevance, Quality, Completeness

Figure 1: **Steps of data measurements framework.** A buyer embeds their data through some embedding model and sends a private query of matrix projections to each seller. Each seller responds with data measurements that allow the buyer to compare and transact with sellers that have the most relevant data.

data breaches, the erosion of privacy, and harmful data misuse. For example, the 2017 Equifax data breach exposed the private records of more than 150 million people [74]. In contrast, decentralized data markets may present a more equitable and efficient approach to data acquisition [53, 55, 36].

On a decentralized marketplace platform, buyers can transact directly with sellers, bypassing inter-mediate data brokers by utilizing decentralized and privacy-enhancing technologies such as smart contracts and trusted execution environments [28, 6]. Bypassing data brokers may result in lower transaction costs and greater market efficiency by allowing data owners to capture more of the revenue generated from their data. In addition, whereas data brokers indiscriminately acquire data and sell bundled datapoints wholesale, data marketplaces could take a more targeted approach to data acquisition. by only paying for the most valuable datapoints, lowering the overall privacy incursion [51]. Lastly, compensating data owners may incentivize greater data access from a more diverse range of individual data producers, which may decrease bias in data acquisition by increasing participation from smaller, more heterogeneous data sources.

However, to fully realize this paradigm shift to decentralized data marketplaces, scalable methods are needed to match buyers with relevant data sellers. A survey of data market participants found that finding relevant sellers was a major source of friction and recommended lowering search costs for the data buyer [36]. In a centralized one-sided marketplace, this process can be facilitated by a data broker. However, in the absence of brokers in a decentralized marketplace, we need federated techniques to signal the value of data sellers to different buyers, each of whom may have different preferences and goals for data acquisition. This problem of seller selection is related to client selection in federated learning [22]. Without new federated methods to lower search costs, market platforms will struggle to attract enough participants to attain the scale and network effects for a sustainable marketplace.

Most current work in data valuation, such as Data Shapley [23], assumes a centralized setting where all data is fully accessible to train models to estimate data value. In a decentralized setting, a seller would not permit data access before payment since data is easily copied. However, a buyer would be reluctant to pay a fair price for data if they cannot be assured of its value. Therefore, a fundamental asymmetry arises between the buyer and seller, related to Arrow's Information Paradox [5], resulting in increased search costs and fewer transactions taking place. New methods must be developed for the decentralized data market setting taking into account only limited, "white-box" data access [10].

To allow a buyer to search for the most promising sellers in a decentralized marketplace, we evaluate *federated data measurements*, which have the advantage of being computationally cheap to compute, task-agnostic, and only require indirect data access. Many different data measurements have been developed to quantify intrinsic, task-agnostic characteristics [48, 40, 43]. Data measurements can be general-purpose, such as central tendency (e.g., mean, median) and "distance" (e.g., Euclidean distance, KL divergence) or modality-specific, such as Fréchet Inception Distance [3] and lexical diversity [31]. Recent work proposed to use conditional diversity and relevance measurements to value data without requiring model training or validation data evaluation [4]. We incorporate their work by evaluating several other definitions of diversity and relevance in the context of private and federated data valuation on medical imaging datasets.

## 3 Federated Data Measurements

Instead of directly attempting to measure the contribution of each datapoint in the seller's dataset, we measure inherent properties of the seller's aggregate dataset through data measurements. These *data measurements*, $\mu$, can be used by the buyer to compare between data sellers. For instance, a seller $j$ with measurement $\mu_j \gg \mu_i$ would be deemed to have more valuable data than seller $i$.

Many data measurements have been developed to quantify intrinsic, task-agnostic characteristics [48, 40, 43]. Data measurements can be general-purpose, such as central tendency and distance metrics, or modality-specific, such as Fréchet Inception Distance [3] and lexical diversity [31]. Many data quality measures have been developed for structured relational data, such as completeness, consistency, and accuracy; however, data quality becomes more complicated for unstructured data [8].

Before measuring the seller's data, a buyer sends a personalized *query*, $\mathbf{Q}$, to each seller. We assume that a buyer has a small sample of reference data, $X_i^{\text{buyer}} \sim \mathcal{D}^{\text{buyer}}$, from the desired distribution to create the query. The buyer communicates this query to the seller, and the seller uses this query to transform their data, calculate the data measurements, and return the measurements to the buyer. The query can be any matrix projection to measure the seller's data. For instance, this basis can be chosen to maximize variance (PCA), independence (ICA), or class separability (LDA) [46, 29, 7]. Empirically, we found PCA with 10 principal directions appropriate for most datasets as most of the variance is captured in the first few components (see Figure 11).

Another common preprocessing step is to embed data into a low-dimensional representation using a deep learning model [47, 42, 69]. The choice of embedding, $f : \mathcal{X} \to \mathbb{R}^d$, can incorporate domain-specific knowledge and has become popular for retrieval augmented generation (RAG) and vector databases [44, 52]. For our benchmark, we use a pretrained CLIP (ViT-16) model — due to its good performance for zero-shot capabilities across a wide range of image domains — to precompute 512-dimensional embedding vectors for each dataset [54]. We envision that more application-specific platforms could use multiple choices of embeddings, such as medical foundation models [49].

First, buyer $i$ sends seller $j$ their query, $\mathbf{Q} = \pi_k\left(f\left(\mathbf{X}^{\text{buyer}}\right)\right)$, where $\pi_k : \mathbb{R}^{n \times d} \to \mathbb{R}^{k \times d}$ computes the first $k$ principal directions using the buyer's reference data. Then, the seller uses this query to transform their data and returns certain information to the buyer to calculate a specified data measurement. The measurement function, $g : \mathbb{R}^{k \times d} \times \mathbb{R}^{k \times d} \to \mathbb{R}$, takes in the projected data from the seller and buyer to produce a scalar data measurement $\mu_{ij} \in \mathbb{R}$, $\mu_{ij} = g\left(\mathbf{Q}\mathbf{C}^{\text{seller}}, \mathbf{Q}\mathbf{C}^{\text{buyer}}\right)$, where $\mathbf{C} \triangleq f(\mathbf{X})^{\top} f(\mathbf{X})$ is the covariance matrix of the embedded data.

In prior work, $g$ has been defined as measuring heuristic notions of *relevance* and *diversity* [4, 70, 21]. For our benchmark, we evaluate the four different definitions of relevance and four definitions of diversity for our decentralized data market setting. Intuitively, relevance should capture the similarity between the buyer and seller. For example, if the buyer has chest X-ray (CXR) images with COVID-19, then a seller with similar COVID-19 CXR images would be more relevant than CXR from normal patients. Likewise, CXR data should be more relevant than MRI data or photography images. We evaluate four definitions of relevance for seller selection.

1. **Negative Euclidean (L2) distance** between the mean vectors of the buyer and seller: $-\left\|\bar{\mathbf{X}}^{\text{buyer}} - \bar{\mathbf{X}}^{\text{seller}}\right\|_2$, where $\bar{\mathbf{X}} \triangleq \frac{1}{k}\sum_{i=1}^{k} \mathbf{Q}_i \mathbf{C}_i$.

2. **Cosine similarity** between mean vectors: $\left(\bar{\mathbf{X}}^{\text{buyer}} \cdot \bar{\mathbf{X}}^{\text{seller}}\right)/\left\|\bar{\mathbf{X}}^{\text{buyer}}\right\|_2 \left\|\bar{\mathbf{X}}^{\text{seller}}\right\|_2$.

3. **Correlation** between mean vectors: $\mathrm{Cov}\left(\bar{\mathbf{X}}^{\text{buyer}}, \bar{\mathbf{X}}^{\text{seller}}\right)/\sqrt{\mathrm{Var}\left(\bar{\mathbf{X}}^{\text{buyer}}\right) \cdot \mathrm{Var}\left(\bar{\mathbf{X}}^{\text{seller}}\right)}$.

4. **Overlap** between principal components [4]: $\sqrt[k]{\prod_{i=1}^{k} \min\left(\lambda_i^{\text{buyer}}, \lambda_i^{\text{seller}}\right)/\max\left(\lambda_i^{\text{buyer}}, \lambda_i^{\text{seller}}\right)}$, where $\lambda_i \triangleq \left\|\mathbf{Q}_i \mathbf{C}_i\right\|_2$ is the magnitude of the projected vector.

For many machine learning applications, using only relevance measures is insufficient to guarantee useful training data. For example, a seller's data may be highly relevant but have duplicate data or imbalanced classes that result in brittle, low-performing models. Intuitively, a seller with X-ray images from 1,000 unique patients contains more non-redundant information than 1,000 X-rays from

a single patient. Then, training on the more diverse seller should lead to better model generalization on unseen test data as more of the input space has been learned [70, 20]. We evaluate four definitions of diversity.

1. **Volume** of the projected covariance [70]: $\log\left(\det\left(\mathbf{QC}^{\text{seller}}\right)\right)$

2. **Vendi score** [21], defined as the exponential of negative entropy of eigenvalues of the covariance: $\exp\left(-\text{trace}\left(\mathbf{QC}^{\text{seller}}\log\mathbf{QC}^{\text{seller}}\right)\right)$.

3. **Dispersion** of the features, measured as the geometric mean of standard deviations [40]:
$$\sqrt[k]{\left(\prod_{i=1}^{k}\text{diag}\left(\mathbf{QC}^{\text{seller}}\mathbf{Q}^{\top}\right)_i\right)}$$

4. **Difference** in the normalized magnitude between principal components [4]:
$\sqrt[k]{\prod_{i=1}^{k}|\lambda_i^{\text{buyer}}-\lambda_i^{\text{seller}}|/\max(\lambda_i^{\text{buyer}},\lambda_i^{\text{seller}})}$, where $\lambda_i \triangleq \|\mathbf{Q}_i\mathbf{C}_i\|_2$.

These data measurements of diversity and relevance are computationally efficient to compute, even for large datasets (>100,000 datapoints), and only require indirect data access from each seller. Additionally, leveraging deep embeddings allows high-dimensional, multi-modal data such as images and text to be measured in a task-agnostic and training-free manner.

# 4 Experiments

**Ranking Sellers with Measurements**    We first evaluate each data measurement in correctly ranking the seller with data IID with the buyer's distribution. For example, when the buyer has reference data from ImageNet, the seller with ImageNet data should have the largest data measurement (see Figure 8). A common metric to evaluate ranking quality in information retrieval is discounted cumulative gain (DCG) [30]. For simplicity, we assume that the IID seller has a maximum gain of 1 and non-IID sellers have a gain of 0. In Table 1, we report the mean rank of the IID seller and DCG over 10 random trials using 20 computer vision datasets (listed in Appendix A). For all experiments, we use 100 datapoints for the buyer query and 10,000 datapoints for each seller unless otherwise specified.

Overall, we find that relevance measurements, such as L2 distance and the "overlap" measure, are better than diversity measurements at ranking the IID seller. This reflects the intuition that relevance directly compares distributional information between buyer and seller. On the other hand, most diversity measures only consider information from the buyer through the query projection step. Among all data measurements, the "difference" measure had the lowest DCG, often ranking the IID seller very low (see Figure 8 for an example).

Table 1: Performance of data measurements for seller ranking

| DATA MEASUREMENT | | AVG. RANKING $\downarrow$ | AVG. DCG $\uparrow$ |
|---|---|---|---|
| RELEVANCE | L2 | $1.25 \pm 0.85$ | $0.94 \pm 0.15$ |
| | COSINE | $1.28 \pm 0.99$ | $0.94 \pm 0.16$ |
| | CORRELATION | $1.34 \pm 1.16$ | $0.93 \pm 0.17$ |
| | OVERLAP [4] | $\mathbf{1.18 \pm 0.53}$ | $\mathbf{0.95 \pm 0.14}$ |
| DIVERSITY | VOLUME [70] | $3.64 \pm 5.28$ | $0.79 \pm 0.30$ |
| | VENDI [21] | $3.38 \pm 2.87$ | $0.69 \pm 0.31$ |
| | DISPERSION [40] | $2.73 \pm 2.87$ | $0.80 \pm 0.29$ |
| | DIFFERENCE [4] | $19.47 \pm 1.04$ | $0.23 \pm 0.0$ |

**Correlation with Downstream Classifier Performance**    Next, we evaluate how useful each data measurement is as a proxy for training data quality. In this experiment, we assume that the buyer wants to use the seller's data to train a model to predict a held-out test set, which is IID with the buyer's query data. We train a model for each seller using their data as a training set and correlate the

Table 2: Correlation test performance across three tasks on four MedMNIST datasets

| PREDICTION TASK | VALUATION METHOD | CORRELATION WITH TEST ACCURACY ↑ | | | | |
| | | BLOOD | ORGAN | PATH | TISSUE | AVG. |
| --- | --- | --- | --- | --- | --- | --- |
| BINARY CLASSIFICATION | L2 | -0.02 | 0.04 | 0.03 | 0.10 | 0.04 |
| | COSINE | 0.16 | 0.09 | 0.13 | 0.20 | 0.15 |
| | CORRELATION | 0.13 | 0.07 | 0.13 | 0.21 | 0.14 |
| | OVERLAP | 0.04 | -0.02 | 0.01 | 0.06 | 0.02 |
| | VOLUME | **0.28** | **0.29** | **0.31** | **0.28** | **0.29** |
| | VENDI | 0.19 | 0.19 | 0.22 | 0.18 | 0.20 |
| | DISPERSION | 0.17 | 0.18 | 0.18 | 0.14 | 0.17 |
| | DIFFERENCE | -0.03 | 0.02 | 0.03 | -0.09 | -0.02 |
| | KNN SHAPLEY | 0.10 | 0.07 | 0.05 | 0.08 | 0.08 |
| | LAVA | -0.02 | -0.02 | 0.02 | 0.01 | 0.00 |
| MULTICLASS CLASSIFICATION | L2 | 0.22 | 0.15 | 0.19 | 0.22 | 0.20 |
| | COSINE | 0.23 | 0.14 | 0.12 | 0.18 | 0.17 |
| | CORRELATION | 0.24 | 0.15 | 0.12 | 0.19 | 0.18 |
| | OVERLAP | 0.27 | 0.19 | 0.19 | 0.24 | 0.22 |
| | VOLUME | **0.42** | **0.35** | **0.32** | **0.36** | **0.36** |
| | VENDI | 0.30 | 0.23 | 0.19 | 0.22 | 0.24 |
| | DISPERSION | 0.22 | 0.20 | 0.12 | 0.18 | 0.18 |
| | DIFFERENCE | -0.23 | -0.14 | -0.14 | -0.18 | -0.17 |
| | KNN SHAPLEY | 0.09 | 0.12 | 0.07 | 0.12 | 0.10 |
| | LAVA | -0.01 | 0.00 | -0.02 | 0.00 | -0.01 |
| K-MEANS CLUSTERING | L2 | 0.22 | 0.23 | 0.20 | 0.19 | 0.21 |
| | COSINE | 0.29 | 0.28 | 0.31 | 0.26 | 0.29 |
| | CORRELATION | 0.29 | 0.29 | 0.31 | 0.26 | 0.29 |
| | OVERLAP | 0.31 | 0.35 | 0.36 | 0.32 | 0.34 |
| | VOLUME | **0.55** | **0.54** | **0.52** | **0.55** | **0.54** |
| | VENDI | 0.45 | 0.45 | 0.49 | 0.48 | 0.47 |
| | DISPERSION | 0.35 | 0.38 | 0.32 | 0.36 | 0.25 |
| | DIFFERENCE | -0.22 | -0.27 | -0.29 | -0.25 | -0.26 |
| | KNN SHAPLEY | 0.01 | 0.05 | 0.02 | -0.01 | 0.02 |
| | LAVA | 0.01 | 0.00 | -0.03 | 0.02 | 0.00 |

resulting model's test performance with the data measurements for that seller. In this way, a seller with a high data measurement value should ideally have test performance for a particular buyer than a seller with a lower data measurement value.

We use four medical imaging datasets (BloodMNIST, OrganMNIST, PathMNIST, and TissueMNIST) from the MedMNIST benchmark (see Figure 6 for example images) [71]. To introduce heterogeneity between sellers, we sample classes from a Dirichlet distribution as standard practice in federated learning to simulate non-IID clients [73, 45]. For each dataset, we evaluate three different prediction task scenarios: binary classification with logistic regression, multiclass classification with a random forest classifier, and K-means clustering. For each data buyer, we randomly sample a subset of classes for multiclass classification and evaluate the accuracy score as the performance metric. For binary classification, we consider the selected subset of classes as "positive" and the other classes as "negative" and evaluate accuracy. For clustering, we set the number of clusters equal to the total number of classes for each dataset and evaluate homogeneity score, a common clustering metric, as the performance metric [58].

For another baseline, we also evaluate two centralized data valuation, KNN Shapley [32] and LAVA [34], using the OpenDataVal framework [33]. We selected these two valuation methods for

their efficient runtime. We split the seller's data into 20% for training and used the other 80% as a validation set. To aggregate a value for each seller, we take the average data value of the validation datapoints. In Table 2, we report these correlations between data measurement and test accuracy for 500 sellers, each with 5,000 datapoints, and average correlations over 10 buyers for each dataset.

Intuitively, we expect that sellers with more similar data as the buyer will learn higher-performing classifiers and be associated with larger data measurement values. For several of the diversity measures (volume, Vendi score), we find a moderate-strong correlation to test performance across datasets and prediction tasks. See Figure 9 for an example of strong correlations between volume measurements and test prediction accuracy. Compared to diversity measures, relevance measures and the centralized data valuation methods (KNN Shapley, LAVA) had a weak correlation with downstream classification performance. These results support that a seller with higher diversity measurements is more likely to have training data that is more useful for a particular, even without specifying the exact prediction task or model architecture. Similar observations between generalization performance and data diversity are reported in determinantal point processes [70, 38].

**Detecting Seller Misreporting with Multiple Queries**   One practical challenge that arises with a decentralized marketplace is ensuring that the seller is not able to "cheat" by artificially inflating the value of their data measurements. In the case of relevance measures, a malicious seller would aim to report mean vectors similar to those of the buyer, but a buyer could avoid sending their own mean vectors to prevent this. However, this strategy would not work for diversity measures, which are independent of the buyer's data given the query.

To counteract this, a buyer could send multiple queries containing "false" directions that may be computed using non-relevant data or even random directions in addition to their actual data (see Figure 10. Then, the buyer could discount sellers with large data measurements in these false directions while only considering sellers with high value using the real query. We evaluate each data measurement's ability to discriminate between data measurements using the real query and false queries with the following ratio

$$\text{ratio}(\%) = \frac{\mu_{\text{real}}}{\text{quantile}(\{\mu_{\text{false}}^{(i)}\}_i^m, \%)}, \tag{1}$$

which is simply the ratio of the data measurement using the real query $\mu_{\text{real}}$ over the %-quantile of measurement using false queries. In our experiment, we compute false queries using 20 non-IID datasets and consider three quantile threshold ratios: 50%, 75%, and 90%. The 50% ratio corresponds to the real IID measurement divided by the median measurement when using buyer queries from the 19 other non-IID datasets.

Table 3: Ratio of measurement using real query over measurements of false queries

| DATA MEASUREMENT | | RATIOS ↑ | | |
|---|---|---|---|---|
| | | 50% | 75% | 90% |
| RELEVANCE | L2 | 1.02× | 0.93× | 0.89× |
| | COSINE | **2.97×** | 1.57× | 1.25× |
| | CORRELATION | 2.83× | 1.53× | 1.18× |
| | OVERLAP | 2.88× | **2.02×** | **1.64×** |
| DIVERSITY | VOLUME | 1.39× | 1.31× | 1.24× |
| | VENDI SCORE | 2.20× | 1.92× | **1.64×** |
| | DISPERSION | 1.91× | 1.73× | 1.58× |
| | DIFFERENCE | 0.38× | 0.30× | 0.27× |

In Table 3, we report measurement ratios and find that most data measurements of relevance and diversity have high ratios, implying that sending multiple queries can be an effective strategy to deal with adversarial sellers that misreport their measurements. This will incentivize the sellers to honestly report their true data measurements as they do not know which queries are real or fake. Sending

additional queries increases communication overhead, but this may be tolerable since each query is cheap — being only a $k \times d$ matrix, where $k \ll n$. For instance, each of our queries is $10 \times 512$ in our experiments.

**Robustness to Duplicate Data**    Because there is no cost to copying data, an adversarial seller may duplicate portions of their data to try to obtain higher measurement values. In Figure 2, we vary the amount of duplicate data to observe the effect on each data measurement when both the seller and buyer have IID data. We note that the implementation of the considered volume method [70] explicitly quantizes the data into a $d$-dimensional hypercube to achieve robustness to duplicate data. Therefore, increasing the amount of duplicated data has a negative effect on volume. For all other data measurements, the value is relatively consistent until falling off for extreme numbers of duplicates, e.g., each datapoint is duplicated 200 times, leaving only $10{,}000/200 = 50$ unique datapoints. Exploring duplicate-robust versions of data measurements would be interesting for future work.

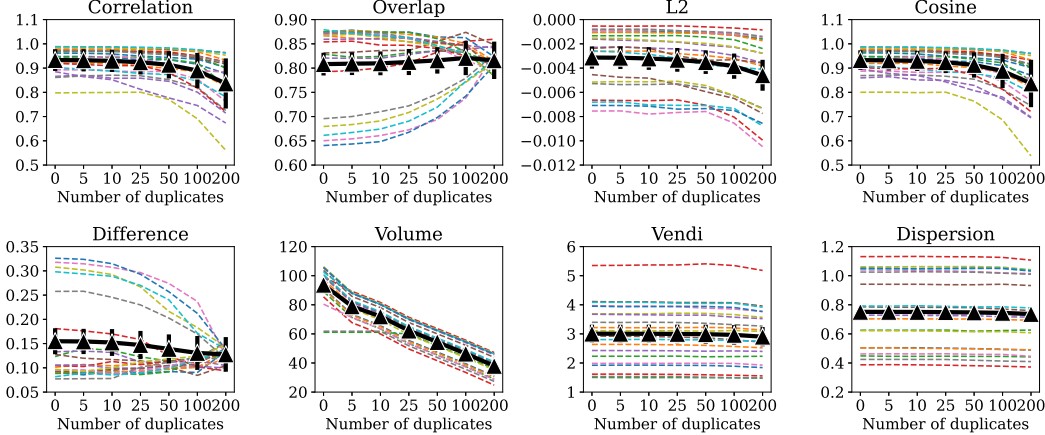

Figure 2: Effect of duplicate data on data measurements. Each seller has 10,000 total datapoints, and a subset of datapoints are duplicated, keeping the total number of datapoints the same. Each colored dotted line represents an individual dataset, and the solid black line represents the average of all datasets. Errors bars represent one standard deviation.

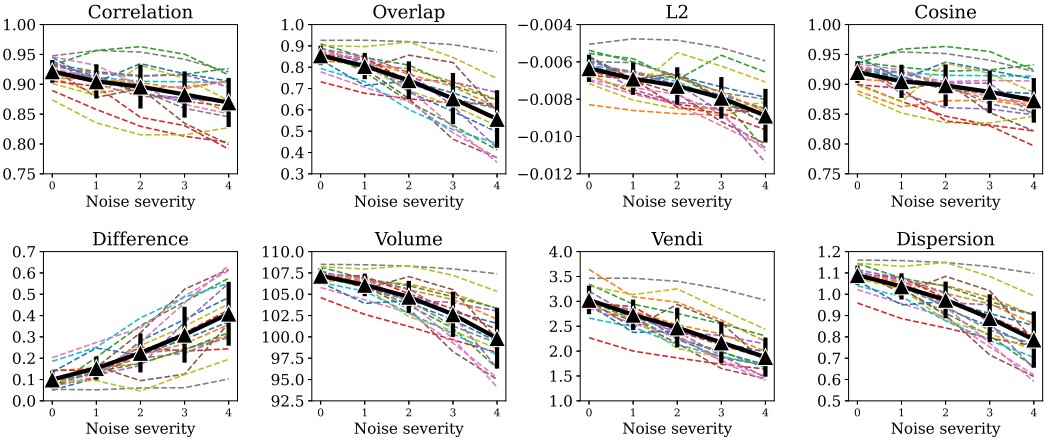

Figure 3: Effect of different types of noise corruptions on each data measurement. See Figure 7 for example images on the ImageNet-C dataset.

**Effect of Noisy and Corrupted Data**    In this experiment, we utilize the ImageNet-C benchmark dataset [26] to study the effect of 19 different types of noise corruptions (blurring, intensity changes,

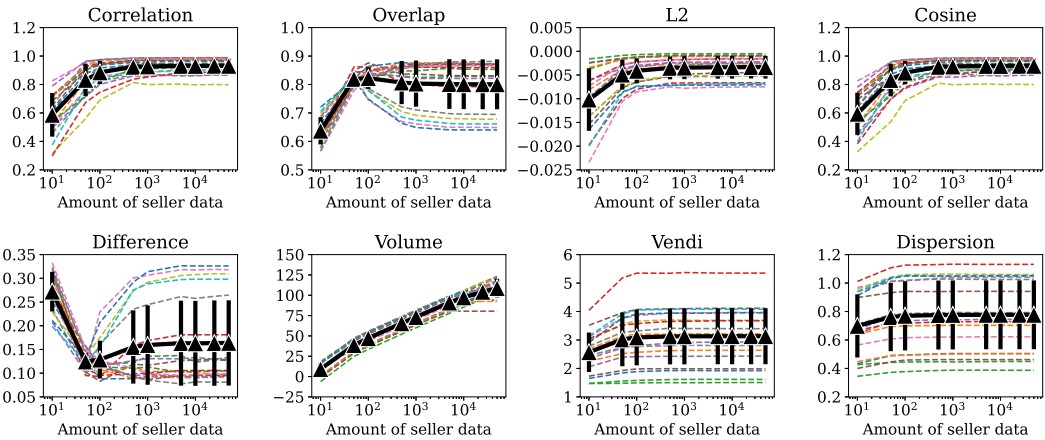

Figure 4: Varying the amount of data each IID seller has while fixing the buyer query to 100 datapoints.

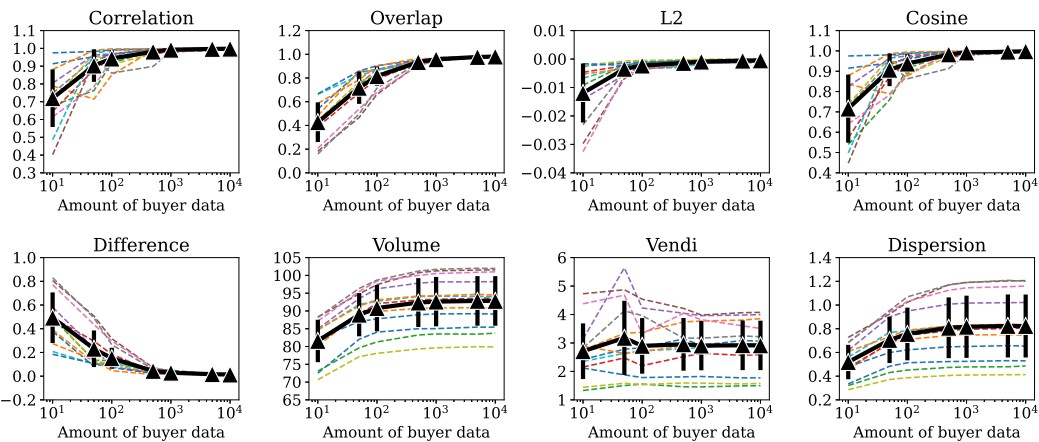

Figure 5: Varying the amount of data in the buyer query has while fixing each seller to 5,000 datapoints.

compression, style effects, etc.) applied to the original ImageNet dataset [59]. Each corruption and noise type has 5 levels of increasing severity. See Figure 7 for an example images. The buyer has 100 datapoints from the original ImageNet dataset, while each seller has 10,000 datapoints from one ImageNet-C corruption type.

As shown in Figure 3, as the severity of the noise/corruption increases, the values of all data measurements decrease (with the exception of the "difference" measurement, which increases). This degradation in diversity and relevance also depends on the type of noise corruption. More subtle changes, such as brightness shifts and saturation, which do not change the spatial information in the image and result in more gradual decreases in measured values. In contrast, heavy corruptions, such as Gaussian noise and glass blur, which affect the image's semantic structure, have much larger effects on measured diversity and relevance.

**Varying the Amount of Seller and Buyer Data**   For these experiments, we use the 20 datasets in Appendix A. In Figure 4, we vary the amount of data each seller has from 10 datapoints to 50,000 datapoints while keeping the buyer's query fixed at 100 datapoints. We find all data measurements, except volume, stabilized after around 1,000 seller datapoints. The volume value continued to increase with the number of seller datapoints. We also vary the amount of in the buyer's query from 10 datapoints to 10,000 datapoints while fixing the number of seller datapoints to 5,000 in

Figure 5. We find that data measurements were relatively stable for most datasets after around 100 query datapoints.

## 5 Discussion

As observed in the experiments, both diversity and relevance measures capture important aspects of data value for a buyer. Relevance measures allow a buyer to filter out irrelevant data and identify sellers with in-domain data distributions. On the other hand, diversity measures, such as volume, reveal which sellers have the most informative and useful data (correlated with test performance, non-duplicated data). As shown with the corruption experiments using ImageNet-C, both diversity and relevance are associated with data quality as noisier and more corrupted data have lower data measurements.

In contrast with prior work [4], we find their "difference" definition of diversity to underperform in most experiments compared to other definitions of diversity. Subjectively, we observe that "difference" measurements tend to be the inverse of "overlap" measurements and thus redundant in terms of information. On the other hand, volume has additional nice properties, such as being robust to data duplication and increasing with the number of seller datapoints. Based on our benchmark experiments, we conclude that cosine similarity and "overlap" are appropriate relevance measures and that the volume-based definition of diversity is well-suited for seller selection.

**Advantages of Federated Data Measurements**    Unlike centralized and training-based approaches to data valuation, using federated data measurements is a lightweight and private way to match a buyer with relevant sellers in a decentralized marketplace with millions of participants. Measuring a seller's data is agnostic to the modeling task and model architecture. This approach allows a buyer to compare the value of multiple sellers relatively without requiring direct access to the seller's data, which would not be allowed before payment. Different choices of embedding functions could be precomputed to serve different types of modalities and domains. In summary, this decentralized data valuation scheme allows private and scalable seller discovery to lower search costs for a data buyer, enabling more efficient markets and lower transaction costs.

**Limitations**    While our work presents an initial benchmark of different data measurements, it is limited in several ways. Firstly, while our data measurements framework can accommodate other types of data modalities such as text and tabular data, we only consider common computer vision datasets for our benchmark. Future work would extend the experiments and embeddings for other domains such as natural language and graphical data. Another limitation is the lack of formal privacy guarantees. While the federated nature of the query and measurement step should prevent reconstruction attacks, techniques such as differential privacy [18] and homomorphic encryption [1] could be employed to provide explicit guarantees. Additionally, further work could incorporate incentive mechanisms to study adversarial seller behavior.

## 6 Conclusion

Reimagining a new decentralized model of data acquisition where individual data producers are fairly compensated for sharing data could help redistribute the economic benefits from AI technology to those whose data enables AI research and development [64]. Decentralized data markets may address issues with current centralized settings by providing a more equitable and efficient exchange of data resources, as well as enabling more collective data governance [53, 17].

In this paper, we presented federated data measurements for decentralized data marketplaces. These measurements allow a buyer to perform seller selection without direct access to the seller's data and are more scalable than current data valuation approaches. We benchmark several properties of data measurements on computer vision datasets and find that a combination of relevance and diversity performs well for several practical data marketplace considerations.

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

# A Datasets

We use the following computer vision datasets in our experiments:

- MNIST Handwritten Digits [41]
- Fashion-MNIST [68]
- EMNIST [12]
- SVHN [50]
- CIFAR10 [37]
- STL-10 [11]
- ImageNet (validation set) [59]
- ImageNet-Sketch [66]
- ImageNet-Rendition [25]
- ImageNet-Adversarial [27]
- ImageNet-V2 [56]
- ImageNet-Corruption [26]
- BloodMNIST (224 by 224 pixel version) from MedMNIST-V2 Benchmark [72]
- BreastMNIST (224 by 224 pixel version) from MedMNIST-V2 Benchmark [72]
- ChestMNIST (224 by 224 pixel version) from MedMNIST-V2 Benchmark [72]
- DermaMNIST (224 by 224 pixel version) from MedMNIST-V2 Benchmark [72]
- OrganAMNIST (224 by 224 pixel version) from MedMNIST-V2 Benchmark [72]
- PathMNIST (224 by 224 pixel version) from MedMNIST-V2 Benchmark [72]
- PneumoniaMNIST (224 by 224 pixel version) from MedMNIST-V2 Benchmark [72]
- RetinaMNIST (224 by 224 pixel version) from MedMNIST-V2 Benchmark [72]
- TissueMNIST (224 by 224 pixel version) from MedMNIST-V2 Benchmark [72]

# B Experimental Setup

Each experiment is averaged over 10 trials of randomly splitting buyer and seller data. For the binary classification task, a random subset of classes was selected for each buyer to be the positive class, while the rest of the classes were labeled negative. For the multiclass classification, a random subset of classes was selected for each buyer, while for the clustering task, all classes were used. Logistic regression was used for the binary task, a random forest model for the multiclass classification, and a K-means model was used for clustering with the number of clusters being initialized to the number of total classes. 100 datapoints were used for the buyer query, and 500 datapoints were used for a test set. For each seller, 5,000 datapoints were randomly sampled from a Dirichlet class distribution and used to train a model to predict the held-out test set. The centralized data valuation baselines (KNN Shapley and LAVA) used 1.000 samples from the seller for training and the rest of the 4000 samples for validation, and the average data value was reported for the seller. The test performance metric was prediction accuracy for binary and multiclass classification, while the homogeneity score was used for the clustering task. In general, the diversity measure is the most correlated with prediction performance across datasets and tasks.

For hardware details, we use an Intel Xeon E5-2620 CPU with 32 cores equipped with Nvidia GTX 1080 Ti GPUs. For baseline implementation of centralized KNN Shapley and LAVA data valuation methods, we use the OpenDataVal package [33] version 1.2.1 with the default hyperparameter settings.

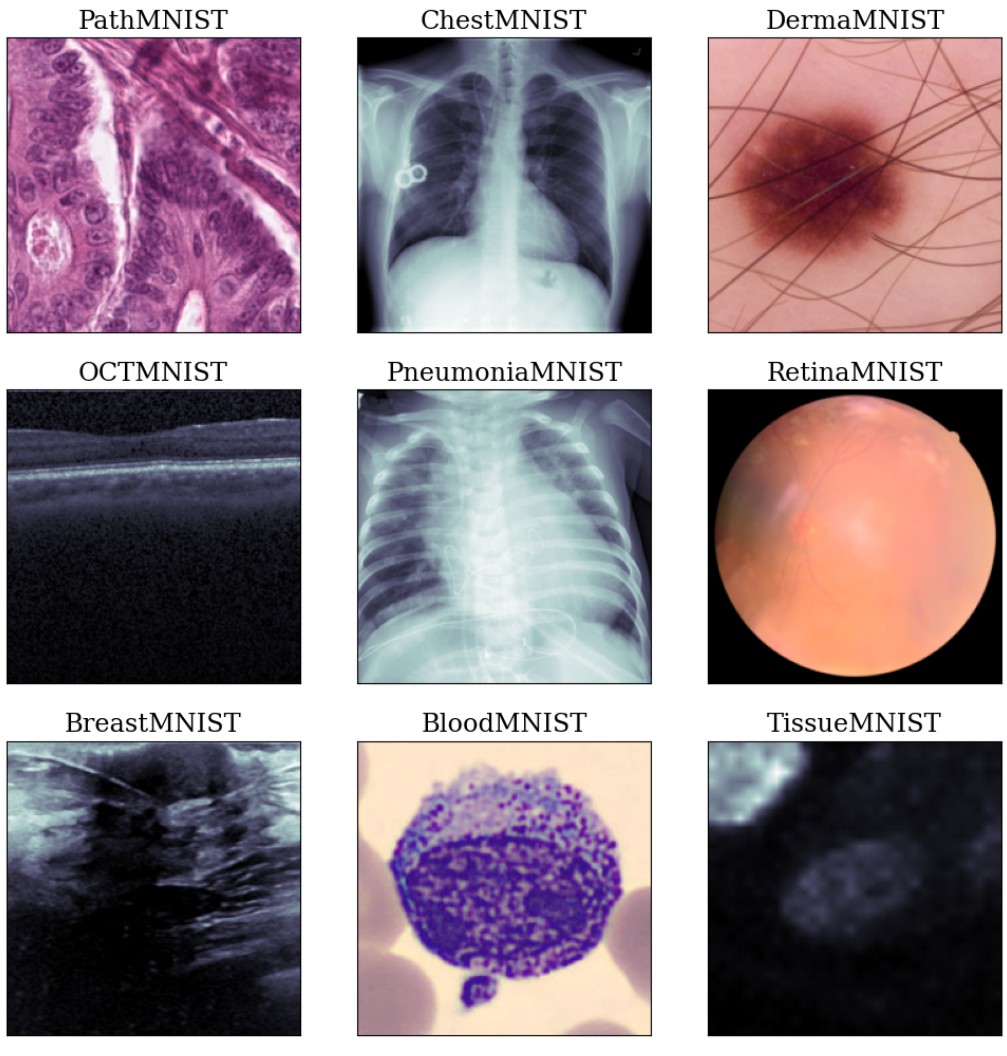

Figure 6: Example images from datasets in the MedMNIST benchmark. See `medmnist.com` for more information.

## C    Additional Figures

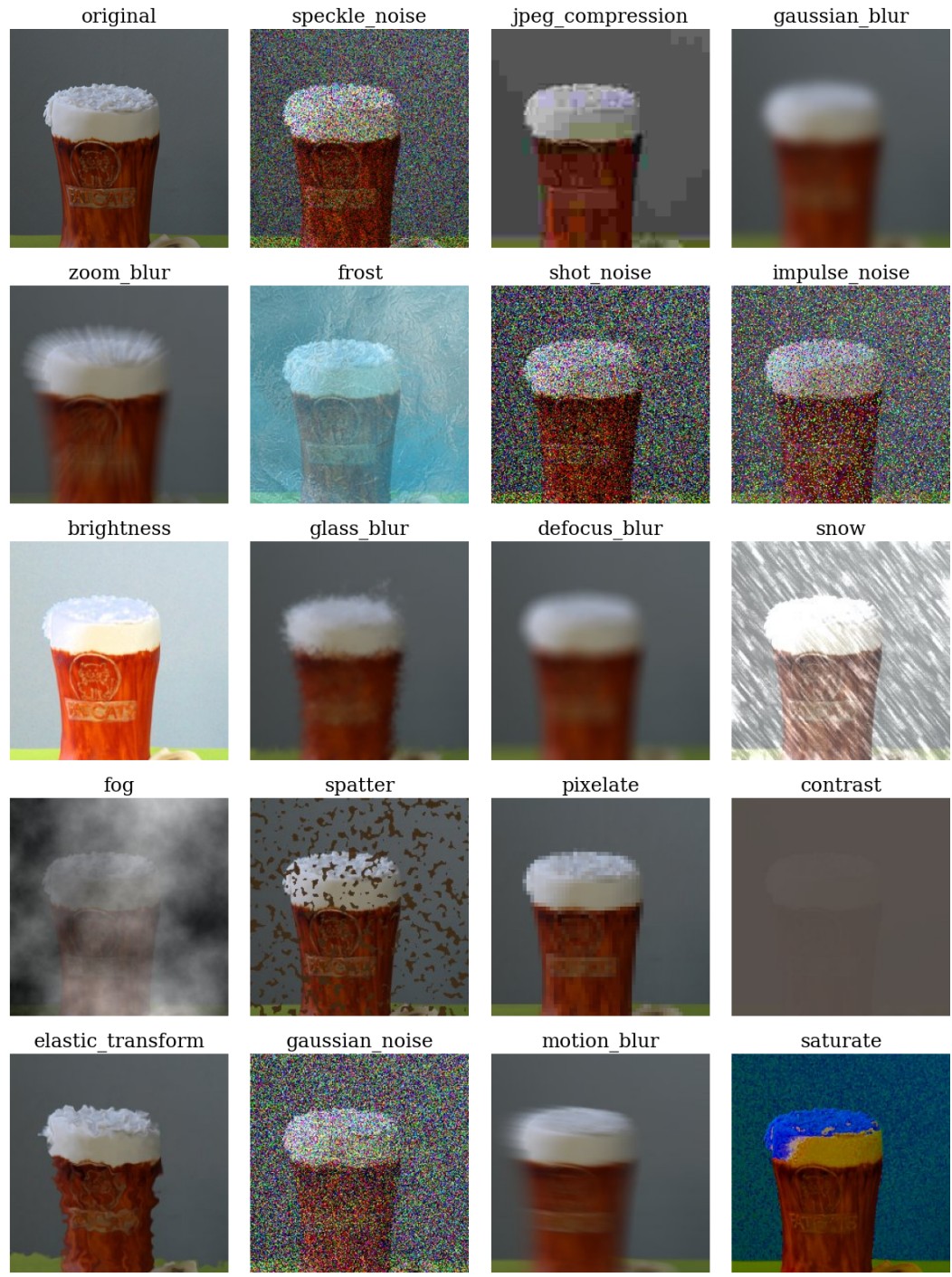

Figure 7: Example noise and image corruptions at the highest severity from the ImageNet-C dataset.

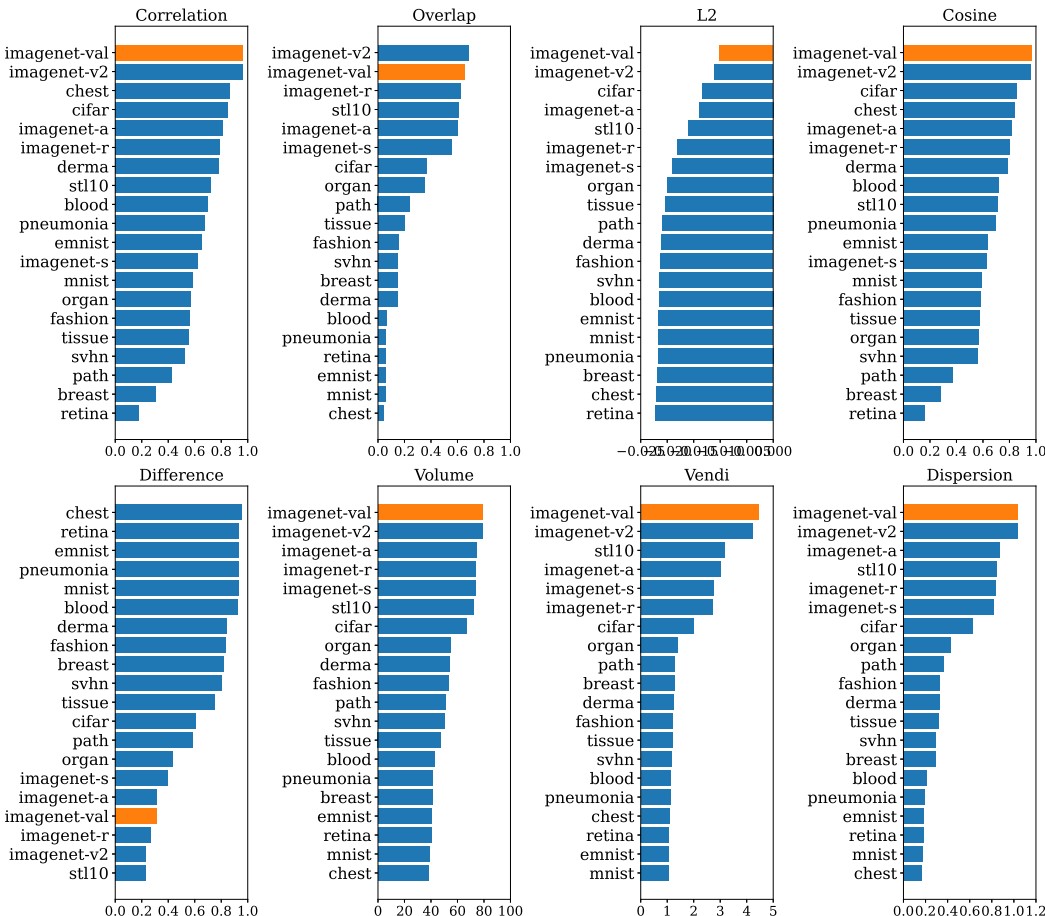

Figure 8: Ranked data measurements of each seller when the buyer query consists of 100 samples from ImageNet. The orange bar denotes the seller with IID data distribution (ImageNet) that should be ranked first.

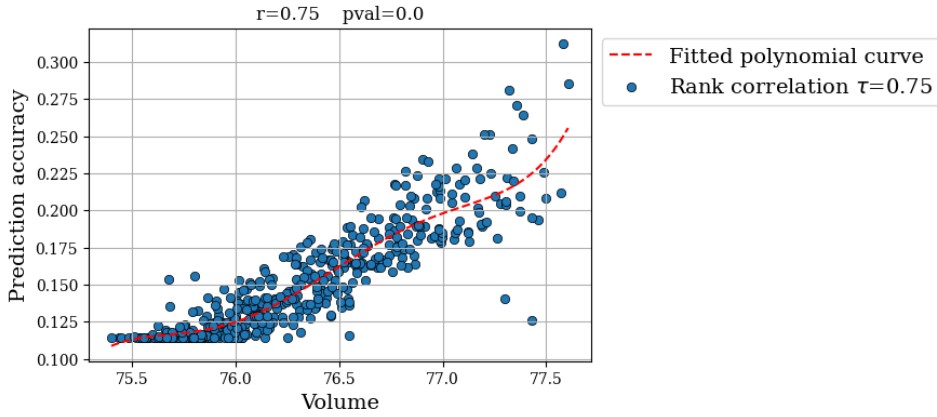

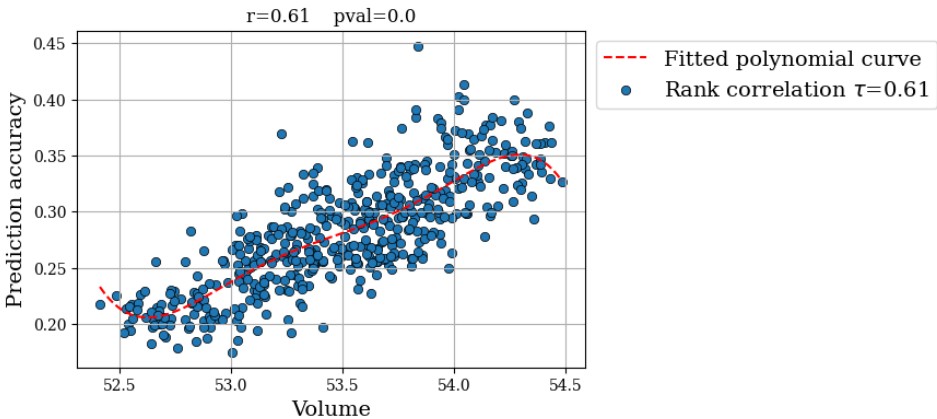

Figure 9: Correlation between volume data measurements and test prediction accuracy on MedMNIST datasets.

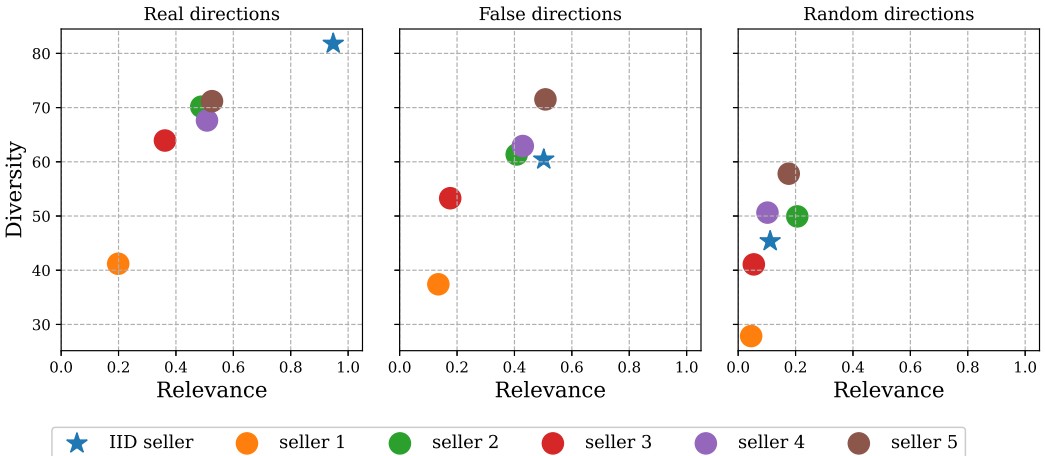

Figure 10: Comparing diversity and relevance measurements when the buyer sends a real query computed on their actual data (left), a false query computed on a random dataset (middle), and a false query computed using random data (right).

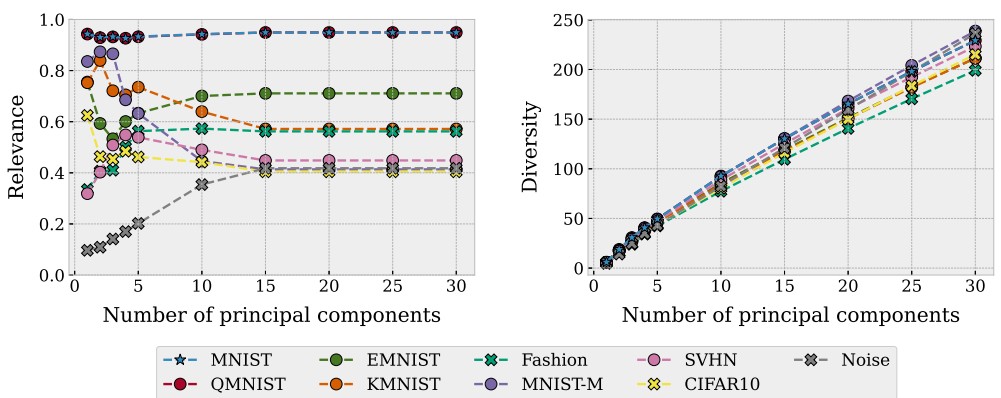

Figure 11: varying the number of principal components used to calculate diversity and relevance. 10,00 samples from the buyer and 10,000 samples from the seller were randomly sampled.

# D  Broader Impact

We believe that AI developers must reconcile important ethical questions regarding data acquisition in current AI development. Class-action lawsuits have been filed against several AI companies for their data collection practices, raising questions about data compensation and consent from data owners. Current data acquisition norms may actively discourage further data sharing, which can hamper the progress and impact of AI, especially in data-limited domains such as healthcare.

Current centralized data brokers acquire data and operate in nontransparent and obfuscatory ways — data is resold between interlinked brokers that make data provenance and traceability of the source difficult [67, 14]. Individuals are often left without recourse or due process over what data is collected or how that data is used [15]. Outdated, incorrect, or out-of-context data may cause harm to the individual. For instance, millions of mugshots of arrested — but not necessarily convicted — individuals are routinely sold on commercial websites and impact those individuals' future employment opportunities and access to housing [39]. Data brokers may also pose risks to civil liberties, such as when individuals' data on race, ethnicity, gender, sexual orientation, immigration status, and other demographic characteristics is utilized in discriminatory practices, policing, and surveillance by corporations and government agencies [62].

In contrast, decentralized data marketplaces may be more robust and transparent. However, to fully realize the promises of a paradigm shift to decentralized data markets, several social, ethical, and technical challenges need to be addressed, such as privacy protections, fair data pricing mechanism, and secure platform infrastructure [63, 19]. Enabling data market platforms also raises ethical concerns and security risks associated with the commodification of personal data, such as the loss of privacy and lack of consent in the collection and use of this data [75]. Marginalized and vulnerable groups are more at risk of data commodification and privacy erosion, and special protections should be enforced for these groups. Safeguards need to be developed to ensure the participation, consent, and compensation of the data owners and producers in establishing the provenance and use of data.

1. Submission introducing new datasets must include the following in the supplementary materials:

    (a) Dataset documentation and intended uses. Recommended documentation frameworks include datasheets for datasets, dataset nutrition labels, data statements for NLP, and accountability frameworks.

    (b) URL to website/platform where the dataset/benchmark can be viewed and downloaded by the reviewers.

    (c) URL to Croissant metadata record documenting the dataset/benchmark available for viewing and downloading by the reviewers. You can create your Croissant metadata using e.g. the Python library available here: https://github.com/mlcommons/croissant

    (d) Author statement that they bear all responsibility in case of violation of rights, etc., and confirmation of the data license.

    (e) Hosting, licensing, and maintenance plan. The choice of hosting platform is yours, as long as you ensure access to the data (possibly through a curated interface) and will provide the necessary maintenance.

2. To ensure accessibility, the supplementary materials for datasets must include the following:

    (a) Links to access the dataset and its metadata. This can be hidden upon submission if the dataset is not yet publicly available but must be added in the camera-ready version. In select cases, e.g when the data can only be released at a later date, this can be added afterward. Simulation environments should link to (open source) code repositories.

    (b) The dataset itself should ideally use an open and widely used data format. Provide a detailed explanation on how the dataset can be read. For simulation environments, use existing frameworks or explain how they can be used.

    (c) Long-term preservation: It must be clear that the dataset will be available for a long time, either by uploading to a data repository or by explaining how the authors themselves will ensure this.

    (d) Explicit license: Authors must choose a license, ideally a CC license for datasets, or an open source license for code (e.g. RL environments).

    (e) Add structured metadata to a dataset's meta-data page using Web standards (like schema.org and DCAT): This allows it to be discovered and organized by anyone. If you use an existing data repository, this is often done automatically.

    (f) Highly recommended: a persistent dereferenceable identifier (e.g. a DOI minted by a data repository or a prefix on identifiers.org) for datasets, or a code repository (e.g. GitHub, GitLab,...) for code. If this is not possible or useful, please explain why.

3. For benchmarks, the supplementary materials must ensure that all results are easily reproducible. Where possible, use a reproducibility framework such as the ML reproducibility checklist, or otherwise guarantee that all results can be easily reproduced, i.e. all necessary datasets, code, and evaluation procedures must be accessible and documented.

4. For papers introducing best practices in creating or curating datasets and benchmarks, the above supplementary materials are not required.

