# OpenReview forum: "Data Measurements for Decentralized Data Markets"
_NeurIPS.cc/2024/Datasets_and_Benchmarks_Track — Submitted to NeurIPS 2024 Track Datasets and Benchmarks_

### Official Review · Reviewer_wcRR · 2024-06-25
**a framework for federated data measurements in decentralized markets**

**Rating:** 9
**Confidence:** 5
**Correctness:** yep
**Clarity:** yep

**Review:**

(S1) The paper studies the problem of data seller selection in decentralized data markets, which is important.
(S2) The paper evaluates and benchmarks various metrics for data measurements of relevance and diversity, providing a comprehensive analysis of their effectiveness.

 (W1) This paper can explore how to combine multiple data measurements of diversity and relevance for data seller selection.

**Strengths:**

pls see review

**Additional Feedback:**

na

**Documentation:**

yep

**Limitations:**

yep

**Opportunities For Improvement:**

(D1) These data measurements of diversity and relevance can be used together for data seller selection. The authors can further explore how different combinations of these measurements impact the effectiveness of seller selection.

**Relation To Prior Work:**

yep

**Summary And Contributions:**

This paper introduces a decentralized data market framework, where data buyers can query data sellers for data relevance and diversity using federated data measurements based on a small data sample. Various federated data measurements of relevance and diversity are evaluated and benchmarked for their effectiveness in seller selection. Experiments include using these measurements to rank data sellers, examining their correlation with the performance of downstream models, and evaluating their robustness against malicious sellers, duplicate or noisy data, and datasets of varying amounts.

---

> ### Author Rebuttal · Authors · 2024-08-12
>
> We greatly thank the reviewer for their close reading of our work, their detailed feedback, and their enthusiastic support!
>
>
> > These data measurements of diversity and relevance can be used together for data seller selection. The authors can further explore how different combinations of these measurements impact the effectiveness of seller selection.
>
> We strongly agree that the strength of both diversity and relevance measurements should be combined for seller selection. For instance, relevance measures can first be used to filter our irrelevant sellers learning a subset of sellers with sufficient similarity to the buyer. A diversity measure can then rank the sellers according to how useful or redundant their data is for the buyer.
>
> In the final paper, we will highlight this point and add additional discussion on combining diversity and relevance measurements for seller selection.

---

### Official Review · Reviewer_2X1C · 2024-07-02
**Good but not enough**

**Rating:** 5
**Confidence:** 4

**Review:**

- __Quality:__ Good. Able to capture the main idea of the framework and the related works in this area. Experiments from different perspectives
- __Clarity:__ Fair. Though it is easy to capture the basic logic of the related area and the purpose of each step in the data measurement process, it is still confusing in many detailed parts. I think it is more because of the way the authors present this paper, which lacks detailed information and explanation. Also, the introduction and abstract need to be revised.
- __Originality:__ Good. Authors build their framework based on the measurement proposed by previous paper, but they do consider different perspectives when applied in decentralized data markets setting, and evaluate them.
- __Significance:__ Good but not enough.

**Strengths:**

the design of experiments includes plentiful perspectives

**Additional Feedback:**

- it could be better to talk more about the motivation, like what is the "seller selection" and why need it, then present the idea of this framework in the introduction
- line 25 typo 'benchmarkiing'
- What are "diversity" and "relevance" mentioned in the introduction? When do we need "diversity"? When need "relevance"? when combined effect?

**Clarity:**

- Not really. A lot of experiment details are missing, so for some part it is difficult to understand the design of the framework. For example, what is the data setting for the experiment "ranking sellers with measurements"?
- Logic relation for background knowledge and problem setting is very confusing. Maybe itemized list or subsection is a good way to present this part.
- Many detailed explanation is not presented. For example, what is "discounted cumulative gain"? How to understand the relationship between DCG and ranking?
- what is the conclusion from experiment "ranking sellers with measurements"? it seems authors only present the experiment results for both 'relevance' and 'diversity' but no very intuitive conclusion and motivation from this experiment.

**Correctness:**

Though authors try to analysis the possible problem in decentralized data markets and use corresponding sythetic data distribution to mimic the settings may exist, a question is, whether such setting makes sense, or what is the data distribution in the real data markets?

**Documentation:**

N/A

**Limitations:**

Though authors try to analysis the possible problem in decentralized data markets and use corresponding sythetic data distribution to mimic the settings may exist, a question is, whether such setting makes sense, or what is the data distribution in the real data markets?

**Opportunities For Improvement:**

- Please revise the abstract and introduction
- Please provide the background knowledge in a more clear logic clue
- Provide more detailed experiment setting, the motivation for experiement and experiment explanation

**Relation To Prior Work:**

Authors build their frameework based on the measurement proposed by previous paper, but they do consider different perspectives when applied in decentralized data markets setting, and evaluate them.

**Summary And Contributions:**

This paper investigates the challenge of seller selection for a data buyer using a framework based on federated data measurements under the setting of decentralized data markets. Authors calculate the several previously proposed measures of data diversity and relevance of several computer vision datasets under a federated setting to evaluate data measurement's ability to rank sellers, correlation with classification performance, and robustness to duplicate and noisy data without requiring the raw data.

---

> ### Author Rebuttal · Authors · 2024-08-12
>
> We thank the reviewer for their detailed comments and hope to address them below.
>
> > A lot of experiment details are missing, so for some part it is difficult to understand the design of the framework. For example, what is the data setting for the experiment "ranking sellers with measurements"?
>
> Thank you for pointing this out! The datasets used for ranking sellers are listed in Appendix A. In the revised paper, we will include more experimental details to improve clarity in the Experiments section as well as the Appendix.
>
> > Logic relation for background knowledge and problem setting is very confusing. Maybe itemized list or subsection is a good way to present this part.
>
> Thank you for your suggestion. In the revised paper, we hope to provide more background on the problem setting using a list of items.
>
> > Many detailed explanation is not presented. For example, what is "discounted cumulative gain"? How to understand the relationship between DCG and ranking?
>
> DCG (https://en.wikipedia.org/wiki/Discounted_cumulative_gain) is a common evaluation metric in information retrieval to measure ranking quality. We will add more detailed explanations of all evaluation metrics used in the experiments in the final paper.
>
> > what is the conclusion from experiment "ranking sellers with measurements"? it seems authors only present the experiment results for both 'relevance' and 'diversity' but no very intuitive conclusion and motivation from this experiment.
>
> From Table 1, we can conclude that the Overlap measurement (and in general relevance measurements over diversity measurements) best distinguishes sellers with the most similar data to the buyer. For example, if the buyer has CIFAR data, a seller with CIFAR data should be ranked higher than another seller with MNIST data. This is what this ranking experiment aimed to evaluate.
>
> > Though authors try to analysis the possible problem in decentralized data markets and use corresponding sythetic data distribution to mimic the settings may exist, a question is, whether such setting makes sense, or what is the data distribution in the real data markets?
>
> We agree that modeling the data distribution of the sellers will be an important consideration in a practical data market. We believe our work represents a first step towards real-world decentralized data markets by studying the seller selection problem with data measurements. Other challenges such as data pricing and distribution shifts in such a market are yet to be studied but will be interesting areas of future work. See [1] for additional considerations of future data markets for AI.
>
> > it could be better to talk more about the motivation, like what is the "seller selection" and why need it, then present the idea of this framework in the introduction
>
> Thank you for this suggestion. In any data market, a key challenge will be to match sellers with buyers. Therefore a subset of sellers will need to be selected for each buyer based on their data needs expressed through queries. We will better motivate the problem of seller selection in the introduction.
>
> > line 25 typo 'benchmarkiing'
>
> Thank you for careful reading. We will fix this.
>
> > What are "diversity" and "relevance" mentioned in the introduction? When do we need "diversity"? When need "relevance"? when combined effect?
>
> The diversity and relevance measurements are described starting from line 104 to line 130 in Section 3. Informally, relevance measures (like [2]) the similarity between the buyer's data and seller's data while diversity measures (like [3]) how informative or non-redundant information is captured in the seller's data (for example many copies of the same datapoint would be less diverse than many unique datapoints in general). In our paper, we find that relevance is more suited to a coarse, first-pass to filter out sellers with irrelevant data (for example filter our non-medical data when the buyer is looking for medical imaging data). On the other hand, diversity measures seem better suited to find the most useful dataset between more closely related sellers. For example, if two sellers both have medical data but one seller has 100 unique patients while the other only has images from 10 unique patients, the diversity of the seller with 100 patients will be higher. We envision that both relevance and diversity measurements will be needed for seller selection and one of the goals of this paper was to find the most useful pair out of 16 different possible combinations of diversity and relevance. Specifically the volume-overlap pair in [2] and [3] seemed to work well for most datasets.
>
> [1] Chen, Lingjiao, et al. "Data acquisition: A new frontier in data-centric AI." arXiv preprint arXiv:2311.13712 (2023).
>
> [2] Amiri, Mohammad Mohammadi, Frederic Berdoz, and Ramesh Raskar. "Fundamentals of task-agnostic data valuation." Proceedings of the AAAI Conference on Artificial Intelligence. Vol. 37. No. 8. 2023.
>
> [3] Xu, Xinyi, et al. "Validation free and replication robust volume-based data valuation." Advances in Neural Information Processing Systems 34 (2021): 10837-10848.

---

> > ### Comment · Reviewer_2X1C · 2024-08-26
> > **I will raise the score**
> >
> > Thank you for the detailed explanation of my previous questions and concerns. I will raise the score to 5

---

> ### Author Rebuttal · Authors · 2024-08-25
>
> We hope we have addressed the reviewer's comments on the writing quality and the presentation of our paper. With the discussion period ending soon, we wanted to see if you had any additional comments or questions. We sincerely thank the reviewer for taking the time to review our work.

---

### Official Review · Reviewer_XpAY · 2024-07-11
**Review for 676**

**Rating:** 4
**Confidence:** 2
**Correctness:** The claims made in the submission cor…
**Clarity:** Yes

**Review:**

Pros:
1. Various measurements are proposed to model the federated data.

2. The experimental results are extensive, with multiple learning tasks, measurements, and scenarios.

Cons:
1. The diversity of datasets can be improved. For example, data from different domains beyond medical images can be included.

2. More insightful discussion of the difference and reliance among different measurements should be given. Why should we use 8 measurements, and are there any criteria to select and use them?

3. What is the motivation for using the federated data framework? As federated learning is not the only candidate in the domain of decentralized learning, the motivation here should be explained.

**Strengths:**

Please see "Pros" in "Review".

**Additional Feedback:**

NA

**Documentation:**

Yes

**Limitations:**

From my perspective, this paper doesn't have potential negative societal impact.

**Opportunities For Improvement:**

Please see "Cons" in "Review".

**Relation To Prior Work:**

Yes

**Summary And Contributions:**

This paper focuses on the scenario of decentralized data market. A benchmark for federated data measurements is proposed. A series of data measurements are investigated in the paper.  Extensive results are provided for analysis.

---

> ### Author Rebuttal · Authors · 2024-08-12
>
> We greatly thank the reviewer for their feedback. We address the comments raised below.
>
> > The diversity of datasets can be improved. For example, data from different domains beyond medical images can be included.
>
> In Appendix A, we list the 21 total datasets used in our experiments, out of which 12 are non-medical imaging. For example, in Figure 7, we show example images from the ImageNet-C dataset [1]. From our experience, we find that our data measurement framework generalizes to tabular and text data, and in the revision, we will include additional results on a wider range of datasets and modalities.
>
> > More insightful discussion of the difference and reliance among different measurements should be given. Why should we use 8 measurements, and are there any criteria to select and use them?
>
> Indeed, one of the aims of our benchmark is to compare the performance between the 8 data measurements. For instance, in Table 2, we see that the Volume measurement is the most correlated with classification performance. Therefore, this measurement would be most useful in scenarios where the buyer is acquiring the seller's data for training. From Table 1, we see that the Overlap measurement is best when ranking sellers with the most similar data to the buyer. Therefore, we recommend using Overlap measurement when comparing sellers with very different data distributions as a course filter and then using the Volume measurement to compare similar relevant sellers for maximum diversity. We will add more discussion on this in the final paper.
>
> > What is the motivation for using the federated data framework? As federated learning is not the only candidate in the domain of decentralized learning, the motivation here should be explained.
>
> To clarify, while federated learning is a related area of decentralized data markets, our work is separate from federated learning as no models are trained. Instead, the data measurements are federated to preserve the privacy of the seller's data before payment (otherwise the buyer could copy the seller's data without paying). We refer to [2] for why federated data acquisition is needed as opposed to current centralized data valuation methods. We will better clarify this point in the intro in the next revision.
>
> [1] Hendrycks, Dan, and Thomas Dietterich. "Benchmarking neural network robustness to common corruptions and perturbations." arXiv preprint arXiv:1903.12261 (2019).
>
> [2] Chen, Lingjiao, et al. "Data acquisition: A new frontier in data-centric AI." arXiv preprint arXiv:2311.13712 (2023).

---

> ### Author Response · Authors · 2024-08-25
>
> We hope we have addressed the reviewer's comments on our paper. With the discussion period ending soon, we wanted to see if you had any additional comments or questions about our worok. We sincerely thank the reviewer for taking the time to review our work.

---

### Official Review · Reviewer_pi4Y · 2024-08-04
**A Good Paper with Extensive Experiments**

**Rating:** 7
**Confidence:** 3
**Correctness:** The paper is technically sound.
**Clarity:** The paper is well-written.

**Review:**

The paper starts by motivating the necessity of decentralized data markets and how by-passing an intermediate data broker would lead to greater market efficiency. I do not necessarily agree with this claim, and believe that having a fully decentralized would incur additional computational costs, e.g., even in this paper's context a buyer need to send a query to multiple seller, leading them to evaluate multiple queries. However, I think that the studying decentralized data markets is relevant, at least from an academic point of view.

The protocol proposed in the paper is relatively simply and I think that there could be room for improvement, especially to reduce the computational burden  on the sellers while evaluating the buyers queries.

The paper executes a sufficient amount of numerical simulations  to evaluate diverse aspects of data quality assessment in decentralized settings.

Overall, the paper provides a relevant and well-executed study. I recommend it to be accepted.

**Strengths:**

* The paper is technically sound and well-executed
* The paper is will-written and well-structured
* The paper introduces a reasonable amount of ideas; federated data measurement, handling adversarial behaviours by the buyers, and robustness to data quality.
* The paper conducts enough experiments to back most of its claim. I believe that the experimental results in this paper could be used by follow-up work.

**Additional Feedback:**

N/A

**Documentation:**

The documentation is decent.

**Limitations:**

The proposed method has a significant computation cost on the sellers side. Moreover, the proposed approach opens the door for a few attacks and adversarial behaviours not covered in the paper. For example one buyer could send multiple queries to sellers with no intention to ever buy anything. This could be an approach to slow-down competitor for example.

**Opportunities For Improvement:**

* I think the authors could provide a more nuanced motivation for decentralized data markets, and could point-out a few of the limitation of this model.
* Handle the computational cost of the proposed approach.
* The proposed approach opens the door for a few attacks and adversarial behaviours not covered in the paper. For example one buyer could send multiple queries to sellers with no intention to ever buy anything. This could be an approach to slow-down competitor for example.

**Relation To Prior Work:**

The paper provides a fair literature review.

**Summary And Contributions:**

The âêr consider the problem of seller selection in decentralized data markets. The paper evaluates different protocols that could be used by buyer to evaluate the quality of sellers data in decentralized data markets. The main protocol is federated data measurements. In this protocol, the buyer sends a private and personalized query based on data embeddings to the seller. The paper evaluates different notion of diversity on multiple datasets and consider differents scenarios and facets of decentralized data markets, including detecting adversarial behaviours, and effects of data volumes.

---

> ### Author Rebuttal · Authors · 2024-08-12
>
> We greatly thank the reviewer for their close reading of our work and detailed feedback!
>
> > motivation for decentralized data markets, and could point-out a few of the limitation
>
> Thank you for pointing this out. In the revised paper, we will incorporate additional motivation and limitations of the decentralized model for data markets.
>
> > computational cost
>
> We agree that maintaining low computation and communication overhead will be crucial in practical decentralized data markets. This is why reason why we focus on data measurements instead of typical data valuation techniques that require model training to avoid computation costs. In follow-up work, we aim to propose solutions for these scalability questions such as maintaining a decentralized directory of sellers so the buyer only sends its query to a subset of sellers instead of all the sellers.
>
> > opens the door for a few attacks and adversarial behaviours
>
> We absolutely agree that a truly practical implementation of our approach for data markets will involve a lot of very exciting research questions on preventing adversarial behaviors such as the DDOS-style attack you mentioned. In follow-up work, we aim to outline various threat models and evaluate potential mitigations through market simulations.
>
> We once again thank the reviewer for their great feedback on our work.

---

### Decision · Program_Chairs · 2024-09-26

**Decision:**

Reject

**Comment:**

The paper proposes 8 metrics to evaluate the usefulness of sellers' data in decentralized data markets.  This is very interesting work.  The empirical evaluation is comprehensive and insightful.  The framework enables buyers to evaluate the usefulness of sellers' data without the sellers having to share any data.  By sending both positive and negative queries, the sellers do not know for which queries they need to return high or low scores, and therefore lying is unlikely to work.  Some reviewers raised concerns about the clarity of the paper.  Those concerns were largely addressed by the rebuttal.  If the paper is accepted, the authors are strongly advised to follow the reviewers' comments to improve the clarity of the paper.